# Computed Tomographic Assessment of Normal Ocular Dimensions and Densities in Cadaveric Horses (*Equus ferus caballus*)

**DOI:** 10.3390/ani15213165

**Published:** 2025-10-31

**Authors:** Maria Luisa Díaz-Bertrana, Lidia Pitti, Ana Sofia Ramírez, Mario Encinoso, Marcos Fumero-Hernández, Inmaculada Morales, Alberto Arencibia, José Raduan Jaber

**Affiliations:** 1Hospital Clínico Veterinario, Facultad de Veterinaria, Universidad de Las Palmas de Gran Canaria, Trasmontaña, 35413 Las Palmas, Spain; luigi.bertrana@ulpgc.es (M.L.D.-B.); inmaculada.morales@ulpgc.es (I.M.); 2Department of Pathology and Food Technology, Faculty of Veterinary Medicine, Universidad de Las Palmas de Gran Canaria, Trasmontaña, 35413 Las Palmas, Spain; anasofia.ramirez@ulpgc.es; 3Hospital Clínico Veterinario, Facultad de Veterinaria, Universidad Autónoma de Barcelona, Carrer de l’Hospital, 08193 Bellaterra, Spain; marcos.vet@outlook.es; 4Departamento de Morfología, Facultad de Veterinaria, Universidad de Las Palmas de Gran Canaria, Trasmontaña, 35413 Las Palmas, Spain; alberto.arencibia@ulpgc.es; 5Grupo de Investigación en Anatomía Aplicada y Herpetopatología, Departamento de Morfología, Universidad de Las Palmas de Gran Canaria, Trasmontaña, 35413 Las Palmas, Spain

**Keywords:** computed tomography, ocular morphometry, head morphometry, horses

## Abstract

**Simple Summary:**

Modern diagnostic imaging techniques, particularly computed tomography (CT), offer valuable information on clinically relevant head structures, including the eyeball, owing to their rapid image acquisition, high spatial resolution, and elimination of superimposed tissues. Comprehensive knowledge of equine ocular dimensions and tissue densities is essential for clinicians and researchers to better understand the biology of the equine eye and its visual capabilities.

**Abstract:**

This study aimed to characterize the computed tomographic (CT) dimensions and contrast attenuation properties of the equine eye. CT scans from 21 horses without ocular abnormalities were analyzed to obtain detailed ocular measurements and attenuation values. In addition, cranial measurements, such as nasal–occipital length and zygomatic width, were incorporated to explore potential anatomical relationships between the skull and intraocular structures. Although most correlations between cranial and ocular parameters were weak, statistically significant associations—particularly those involving lens dimensions and anterior chamber measurements—suggest that skull morphology may exert a subtle influence on ocular anatomy.

## 1. Introduction

The equine eye is among the largest of all terrestrial mammals and plays a pivotal role in the daily activities of horses. Unlike cattle, horses are extensively used in sports, leisure, and fieldwork, relying heavily on vision for both performance and safety [1,2]. Anatomically, the equine globe is distinguished by its considerable axial length and broad visual field, adaptations that facilitate early detection of predators and environmental hazards [3]. However, these same structural features also pose clinical challenges, as the eye is predisposed to trauma and ocular disease, which can markedly compromise vision and, consequently, the animal’s welfare and utility [4,5].

Ophthalmic conditions in horses are increasingly recognized in veterinary practice. The growing awareness of equine eye health has led to a higher demand for routine ophthalmologic examinations and has consolidated equine ophthalmology as a key discipline within veterinary medicine, where up-to-date knowledge and advanced diagnostic tools are essential [1]. Advances in imaging technology and therapeutic options have markedly improved the prognosis of many ocular diseases; nonetheless, continuous training remains critical for veterinarians to effectively diagnose and manage these conditions.

Diagnostic imaging has become a cornerstone of equine ocular assessment. Ultrasonography is widely used to evaluate diverse ocular structures, including the anterior chamber, iris, ciliary body, lens, vitreous body, retina, sclera, optic disc and optic nerve [3]. More advanced modalities, such as magnetic resonance imaging (MRI) and computed tomography (CT), provide complementary information and enable more comprehensive diagnostic evaluations. MRI offers superior soft tissue imaging and contrast resolution, whereas CT provides rapid acquisition and detailed assessment of orbital bone structures [4]. Consequently, CT is particularly helpful for characterizing variations in ocular dimensions, supporting accurate diagnosis, treatment planning, and surgical decision-making [5]. Although techniques such as ultrasonography, ophthalmometry, and retinoscopy have been used to measure the equine eye [6], these methods alone cannot provide all the information necessary for precise diagnosis and therapeutic planning.

In recent years, CT has been increasingly employed to investigate ocular structures in a variety of animal species, including horses [6], dogs [7], cats [8], sea turtles [9], puffins [10], and even extinct taxa [11]. In horses, previous studies have reported CT-based measurements of ocular components across different breeds, establishing reference values relative to body weight [5]. However, these studies have not explored potential associations between ocular dimensions and visual function, nor have they considered cranial morphology as a possible contributing factor. Establishing normative data on ocular dimensions and their relationship with cranial morphology is essential not only for improving diagnostic accuracy and surgical planning in equine ophthalmology but also for advancing our understanding of the functional characteristics and limitations of equine vision [6,7,8]. Therefore, the present study sought to provide a comprehensive CT characterization of the normal equine eye by establishing reference values for ocular dimensions, volumes, and radiodensities, and by evaluating potential correlations with cranial morphometrics.

## 2. Materials and Methods

### 2.1. Animals

The study population consisted of twenty-one equine cadavers of different ages, all of which had been treated at the Equine Medicine and Surgery Service of the Veterinary Teaching Hospital, University of Las Palmas de Gran Canaria. Each animal had undergone a complete ophthalmologic examination by a certified ophthalmologist as part of its clinical evaluation, and only those confirmed to be free of ocular abnormalities were included in this study. Animals were excluded if they presented any history, clinical, or radiological evidence of ocular or retrobulbar disease. All horses were euthanized for reasons unrelated to this research. The most frequent causes of euthanasia were colic, laminitis, and, less commonly, fractures. Immediately after euthanasia, the carcasses were donated for research purposes and stored in a cold chamber at −20 °C to preserve soft tissues and prevent autolytic changes. Once completely frozen, the heads were separated using an electric saw and maintained under frozen conditions until imaging. This protocol was particularly important given the high ambient temperatures in the Canary Islands, where any delay between death and freezing could compromise tissue integrity. As the study was conducted exclusively on cadaveric specimens, approval from the institutional animal welfare committee was not required. Informed consent was obtained from the owners, who were advised that all data derived from the animals would be handled confidentially, in accordance with applicable regulations (Spanish Royal Decree 53/2013, which regulates the protection of animals used for scientific purposes, and Directive 2010/63/EU of the European Parliament on the same subject), and used solely for research or educational purposes.

### 2.2. CT Technique

CT examinations were performed on previously frozen specimens after a 12 h thawing period at room temperature. Images were obtained using a 16-slice helical CT scanner (Toshiba Astelion, Canon Medical Systems, Tokyo, Japan). Only the heads were scanned, and all external tissues, including the skin, muscles, and eyelids, were preserved intact to maintain the normal anatomical configuration of the orbital and periocular structures during imaging. Each head was positioned symmetrically in a prone, rostrocaudal orientation, and scanned following a standard clinical scanning protocol (120 kVp, 80 mA, 512 × 512 acquisition matrix, 1809 × 858 field of view, pitch 0.94, and a gantry rotation of 1.5 s). Images were acquired with a 0.6 mm slice thickness in dorsal, transverse, and sagittal planes, using both bone and soft-tissue reconstruction algorithms. The resulting datasets were subsequently imported into an image viewer (OsiriX MD, Apple, Cupertino, CA, USA) for visualization, manipulation, and morphometric analysis of the eyeball and orbit.

### 2.3. Measurements

Manual contouring on every slice of the CT images was performed by a single examiner, with contours for the eyeball and the lens generated for both eyes in each patient. Measurements for the intraocular distances were calculated by the same single examiner in all patients. The measurement approach followed the protocol outlined in previous studies conducted on dogs and cats [7,8], the loggerhead turtle [9], the puffin [10], and horses [5]. Cranial measurements, including head length, head width, and orbital depth, were performed [12]. Additionally, both eyes (*n* = 40) were evaluated using CT images reconstructed in oblique, sagittal, transverse, and dorsal planes for all animals. Image assessment was carried out using a soft tissue window setting to enhance the contrast of non-osseous structures. The specific parameters and dimensions obtained are detailed below:Eyeball equatorial width and height: Maximal anterior–posterior distance (Figure 1A) and maximal lateromedial distance of the eyeball, measured perpendicular to the axial length (Figure 2B).Orbital cavity height: Maximal dorsoventral distance of the orbital cavity at the level of the eyeball.Lens size: Maximum anterior–posterior distance (axial length) (Figure 1B) and lateromedial distance (equatorial width), measured at the widest dimension (Figure 2A).Anterior chamber, posterior chamber, and lens densities: Attenuation values measured by placing regions of interest (ROIs) centrally in each structure (Figure 3).

Eyeball rostrocaudal length: Maximal anteroposterior distance of the eyeball from the internal surface of the cornea to the internal surface of the choroid/retina/sclera (Figure 4A).Orbital cavity rostrocaudal length: Maximal anteroposterior distance of the orbital cavity measured at the level of the eyeball.Lens rostrocaudal length: Maximum anteroposterior distance of the lens measured along its midline (Figure 4B).Anterior and posterior chambers width: Maximal lateromedial distance across the anterior and posterior chambers (Figure 5).

**Figure 4 animals-15-03165-f004:**
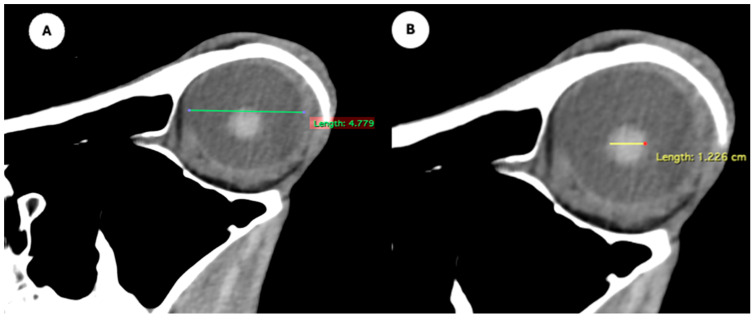
Parasagittal multiplanar reconstruction (MPR) images of the equine eyeball showing (**A**) eyeball length and (**B**) the maximal antero-posterior distance of the lens measured along its midline.

**Figure 5 animals-15-03165-f005:**
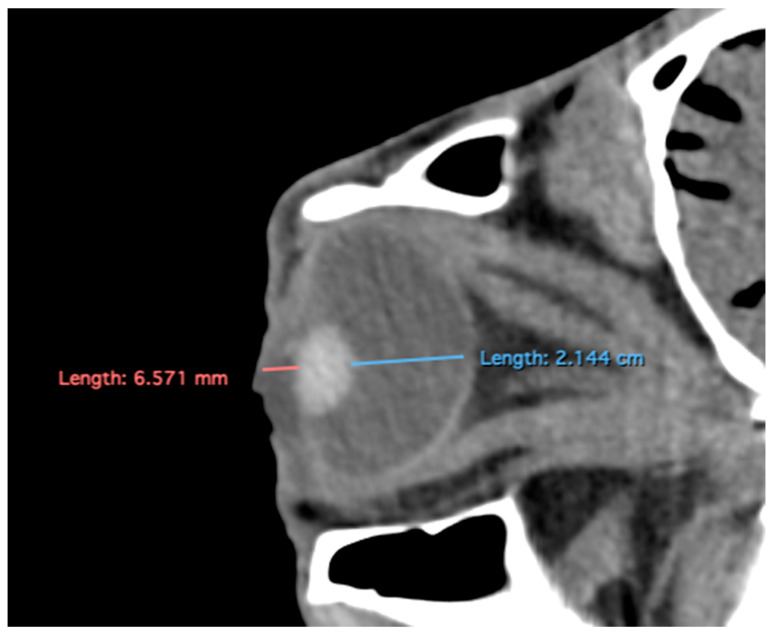
Transverse multiplanar reconstruction (MPR) image showing the widths of the anterior and posterior chambers, represented by the maximal latero-medial distance across each chamber.

The study also evaluated cranial dimensions by measuring skull length—from the occipital protuberance to the rostral border of the nasal bone (Figure 6A)—and zygomatic width, defined as the distance between zygomatic bone borders (Figure 6B) [12].

### 2.4. Statistical Analysis

Statistical analysis was performed using IBM SPSS Statistics (Statistical Package for the Social Sciences version 29, Chicago, IL, USA). The data were described using descriptive statistics such as mean, median, range and standard deviation (SD), while data distribution was assessed using a Shapiro–Wilk test for normality. The Mann–Whitney U test was used to compare measurements between the right and left eye. Correlations between various eyeball measurements and other quantitative variables were assessed by calculating Spearman’s Rank correlation. The level of significance was set at a *p*-value of less than 0.05.

## 3. Results

Descriptive statistics for the study population are summarized as follows. Regarding breed distribution, 20 animals (95.23%) were crossbred, and one (4.76%) was a pure English Horse. In terms of sex, 18 individuals (85.71%) were male and 3 (14.29%) were female. The average body weight was 467.54 ± 34.94 kg (range: 400–530 kg), and the mean age was 17.00 ± 4.99 years (range: 10–25 years). Regarding head measurements, the mean nasal–occipital length was 47.80 ± 2.00 cm (range: 46.5–53.0 cm), and the mean zygomatic width was 17.15 ± 1.25 cm (range: 16.0–20.0 cm).

Table 1 provides summary statistics (mean, median, range, and standard deviation) for the internal ocular measurements of the left eye, right eye, and both eyes combined across all 20 horses. The average ocular measurements indicated that the eyeball height (5.19 ± 0.13 cm) was greater than its width (4.28 ± 0.12 cm), confirming the characteristic oval shape of the equine eye. The mean axial length was 5.09 ± 0.17 cm, consistent with previous descriptions of large-globe morphology in horses. The orbital cavity measured 5.27 ± 0.12 cm in height and 6.29 ± 0.14 cm in length. The anterior and posterior chambers had mean widths of 0.56 ± 0.07 cm and 2.26 ± 0.09 cm, respectively. The lens exhibited a mean height of 1.53 ± 0.06 cm, width of 1.23 ± 0.06 cm, and length of 1.63 ± 0.04 cm. These results collectively reflect the regular proportions and bilateral symmetry of the equine eyeball and its internal structures, as detailed in Table 1. The Mann–Whitney U test revealed no statistically significant differences in the measurements between the right and left eyes. Similarly, when all variables were considered collectively, the analysis confirmed the absence of significant differences between both eyes

Following a Shapiro–Wilk test, approximately half of the variables were found to deviate from a normal distribution. Consequently, the non-parametric Spearman correlation test was employed to assess relationships among morphological measurements. Thus, four of the eleven measurements showed significant weak correlations with head measurements. Nasal–occipital length showed a positive correlation with zygomatic width, and weak negative correlations with lens height and lens width. Zygomatic width also showed a weak but statistically significant negative correlation with posterior chamber width. No other significant correlations were found between skull measurements and the dimensions of the eyeballs, orbital cavities, lens size, or anterior–posterior chamber widths. Age showed a statistically significant correlation only with zygomatic width (ρ = −0.516, *p* < 0.001), indicating a moderate negative association with high statistical significance. Additionally, two variables exhibited moderate-to-weak correlations with body weight: nasal–occipital length (ρ = 0.429, *p* = 0.005) and lens width (ρ = −0.370, *p* = 0.016).

All eyeball measurements were intercorrelated. Weak correlations were observed between length and width (ρ = 0.329; *p* = 0.033) and between length and height (ρ = 0.371; *p* = 0.015). A strong correlation was found between width and height (ρ = 0.704; *p* < 0.001). Orbital cavity height and length were significantly correlated (ρ = 0.639; *p* < 0.001), indicating a strong association. Orbital cavity height also showed significant correlations with all eyeball measurements: width (ρ = 0.360; *p* = 0.019), height (ρ = 0.419; *p* = 0.006), and length (ρ = 0.542; *p* < 0.001). In contrast, orbital cavity length was only correlated with eyeball length (ρ = 0.422; *p* = 0.005). Lens dimensions showed varying degrees of correlation. A strong correlation was found between lens height and lens width (ρ = 0.788; *p* < 0.001), while a weak correlation was observed between lens height and lens length (ρ = 0.306; *p* = 0.048). No significant correlation was found between lens width and lens length (ρ = 0.279; *p* = 0.074). Lens width demonstrated moderate-to-weak correlations with several eyeball and orbital cavity measurements: eyeball width (ρ = 0.346; *p* = 0.025), eyeball height (ρ = 0.332; *p* = 0.032), eyeball length (ρ = 0.523; *p* < 0.001), orbital cavity height (ρ = 0.349; *p* = 0.023), and orbital cavity length (ρ = 0.484; *p* = 0.001). Lens height showed moderate correlations with eyeball length (ρ = 0.538; *p* < 0.001), orbital cavity height (ρ = 0.435; *p* = 0.004), and orbital cavity length (ρ = 0.539; *p* < 0.001). Conversely, lens length was not significantly correlated with any eyeball or orbital cavity measurements. Finally, the widths of the posterior and anterior chambers were moderately correlated (ρ = 0.580; *p* < 0.001). Both variables showed weak to moderate correlations with all eyeball, orbital cavity, and lens dimensions (ρ = 0.337–0.650; *p* = 0.029 to <0.001), except for the posterior chamber width with eyeball length (ρ = 0.232; *p* = 0.140), and the anterior chamber width with lens length (ρ = 0.204; *p* = 0.195), which were not statistically significant.

Related to attenuation, the ROI (regions of interest) for measuring the attenuations of the anterior chamber, posterior chamber and lens densities are shown with different color circles (illustrated in Figure 3). The mean lens attenuation was 124.93 Hounsfield units (HU) with a range of 101–147 HU. The posterior chamber exhibited an average attenuation of −4.26 HU (range: −8 to −2 HU), while the anterior chamber displayed an attenuation of −8.05 HU (range: −14 to −4 HU).

## 4. Discussion

Modern diagnostic imaging modalities, including ultrasonography, CT and MRI, have transformed the evaluation of ocular structures in veterinary medicine [1,2,3,4,5,6,7]. These techniques provide multiplanar image acquisition with high anatomical resolution, excellent tissue contrast, and elimination of superimposition artifacts [1,2,8,9,10]. Consequently, they have significantly improved the visualization and diagnosis of ocular and periocular structures, enabling accurate detection and characterization of a broad range of pathologies [2,4,5]. MRI offers superior soft tissue contrast and true multiplanar imaging without ionizing radiation, but it requires general anesthesia and considerably longer acquisition times compared to CT [7]. By contrast, CT enables rapid image acquisition, three-dimensional reconstructions, and high-quality visualization of both intra- and extraocular structures. In addition, it provides reliable anatomical measurements and detailed assessment of orbital morphology [7].

To the best of our knowledge, this is the first study to report quantitative measurements of normal equine cadaveric eyes while also assessing potential associations between ocular and cranial dimensions. Available data on this topic in clinically normal horses remain scarce [5,13], with previous studies primarily focused on comparing ocular CT measurements across breeds or correlating them with body weight. Nonetheless, these investigations presented some limitations related to their retrospective design and the absence of ophthalmologic examinations. In addition, Hollis et al. (2019) [5], reported computed tomographic ocular dimensions only performed in one eye, which may have provided incomplete information, as studies in other species, including dogs [7], cats [8], sea turtles [9], puffins [10], extinct taxa [11], and Jeju horses [13], have typically evaluated both eyes.

The CT images obtained in this study provided adequate visualization of orbital soft tissues and ocular adnexa in horses, with high spatial and moderate contrast resolution. Although some studies have employed contrast media to enhance the visualization of intraocular components such as the aqueous and vitreous humors [7,8], this was unnecessary here, as the sclera, extraocular muscles, ciliary body, lens, and vitreous chamber were sufficiently delineated without enhancement. This allowed the identification of most intraocular structures previously reported in other species, including reptiles such as the Komodo dragon [14,15], green iguana, common tegu, and bearded dragon [16], as well as the rhinoceros iguana [17], and several chelonian species inhabiting marine, freshwater, and terrestrial environments, such as the green turtle, black-bellied slider, loggerhead turtle, and red-footed tortoise [9,17,18]. Comparable findings have also been reported in domestic mammals, including dogs [7] cats [8], and horses [5,13].

Previous studies in horses have focused almost exclusively on adult specimens [5,13], whereas our work included a broader age range, by incorporating a foal. This inclusion provides preliminary insights into potential age-related anatomical variation, although larger juvenile cohorts are needed to draw definitive conclusions. Some investigations have also examined associations between ocular dimensions and body weight, reporting significant positive correlations for most ocular parameters except the lens and the anteroposterior diameter of the anterior chamber [5]. However, these correlations were only fair to moderate, suggesting that ocular dimensions cannot be reliably predicted based solely on body weight.

In contrast, our study assessed the relationship between ocular anatomy and cranial morphometrics—specifically nasal–occipital length and zygomatic width—thus providing a skeletal-based morphometric perspective. Interestingly, although the correlations between skull dimensions and certain intraocular structures (such as lens height and posterior chamber width) were weak, they were statistically significant. These findings align with other investigation conducted in horses [5], where lens length did not correlate significantly with skull dimensions or other ocular parameters. By contrast, we identified strong internal correlations among lens dimensions (height and width), which may have practical implications for surgical planning and prosthetic design. Overall, our results suggest that cranial morphology exerts subtle but measurables influences on ocular anatomy, highlighting the need for further investigations with larger and more diverse populations to validate and expand these preliminary associations.

Beyond examining the relationship between cranial and ocular structures, our study also provides a biomechanical perspective on equine visual anatomy. Horses possess some of the largest eyes among terrestrial mammals; nonetheless, their visual acuity remains relatively low—typically ranging between 20/30 and 20/60—requiring them to be much closer to an object than humans to perceive similar detail [19,20,21]. This limited resolution has been attributed to the relatively low density of retinal ganglion cells, especially within the narrow visual streak, which serves as the main area of visual focus in the equine retina [21,22,23,24,25]. Supporting this, previous studies have demonstrated that even the zone of highest retinal acuity in horses achieves only ~16.5 cycles per degree, with peripheral areas dropping to ~2.7, which challenges the assumption that head movement can compensate for limited ocular resolution [19,20]. By contrast, species with higher retinal acuity tend to exhibit stronger correlations between cranial dimensions and ocular metrics [9,10,11,18,23]. In our study, we observed weak but statistically significant associations between cranial parameters and ocular metrics such as lens height and posterior chamber width. This pattern mirrors findings in other species where low correlations are linked to limited visual acuity [18,23]. Taken together, these results suggest that while equine ocular geometry is constrained by cranial morphology, such constraints do not enhance visual resolution, thereby underscoring an inherent functional limitation in the equine visual system.

From a practical perspective, the morphometric data generated in this study may contribute to improved diagnostic accuracy of ocular pathologies and support pre-surgical planning for orbital and intraocular procedures. Importantly, the precise anatomical measurements obtained here could serve as a foundation for the development of ocular prostheses following enucleation and for designing intraocular lens implants in horses, areas where reliable reference values are still scarce. Ultimately, this line of research aims to provide equine practitioners with imaging-based guidelines that not only aid diagnosis and prognosis but also innovative therapeutic and surgical approaches, thereby enhancing patient welfare and clinical outcomes.

Although similar evaluations have previously been documented in horses [13], our results indicate that, despite differences in composition, the aqueous and vitreous humors displayed subjectively similar radiodensities, both being hypodense compared to the lens. These findings are consistent with observations in other species [7,8]. Nonetheless, studies in dogs and cats have reported significant differences between the densities of both humors, as well as marked enhancement following contrast administration. It is important to highlight that changes in the radiodensity of vitreous or aqueous humors may reflect underlying conditions such as hemorrhage, inflammation, degeneration, clot formation or neoplasia. Such variations could therefore be clinically relevant for determining the etiology of ocular abnormalities. Given that the present investigation was conducted using cadaveric specimens, further studies in live animals are required to evaluate the clinical applicability of these findings.

Although the present work focused exclusively on the characterization of normal ocular anatomy, the reference values and structural relationships described here may serve as a basis for interpreting pathological changes in clinical settings. Alterations in ocular dimensions or radiodensity could be associated with conditions such as intraocular neoplasms, inflammatory or degenerative processes, or post-traumatic remodeling of orbital structures. In this context, the present morphometric data may support early recognition of abnormal patterns in computed tomographic examinations of horses with ocular or periocular disease. Future studies incorporating clinical and pathological cases are warranted to validate these anatomical benchmarks and to assess how different disease processes may influence ocular geometry and internal structure.

The limited sample size represents a primary limitation of this study. However, animals were deliberately selected following ophthalmological examinations to confirm the absence of ocular, retrobulbar, or adjacent structural pathology that might affect measurements. This careful selection was intended to establish accurate and reliable CT reference values for normal ocular dimensions in horses. While post-mortem imaging eliminates motion artifacts and avoids the need for anesthesia, it may also introduce subtle alterations in tissue hydration, intraocular pressure, or anatomical integrity—factors known to affect ocular dimensions in cadaveric specimens compared to live **animals (e.g., lens** position shifts in porcine eyes approximately six days postmortem) [26,27,28]. Despite these limitations, post-mortem CT remains a valid and practical tool for anatomical characterization, particularly when ethical or logistical constraints preclude in vivo imaging of healthy, anesthetized horses. Moreover, comparative studies in other **species (e.g., porcine** cardiac DTI) have shown high concordance between in vivo and post-mortem quantitative imaging parameters [28], supporting the reliability of ex vivo measurements under controlled conditions [29,30,31,32].

## 5. Conclusions

This study provides a detailed CT characterization of normal ocular dimensions and radiodensities in cadaveric horses, together with a preliminary analysis of their correlations with cranial morphometrics. The reference values established here for ocular and intraocular structures offer valuable baseline data for future diagnostic, clinical and surgical applications in equine ophthalmology.

Although the statistically significant associations observed between skull dimensions (nasal–occipital length and zygomatic width) and certain intraocular **parameters (e.g., lens** height, posterior chamber width) were weak, they suggest that cranial morphology may exert subtle influences on ocular anatomy. Consistent with findings in other species with limited visual acuity, these weak correlations suggest that external morphometry alone is insufficient to accurately predict intraocular measurements.

These results align with the known modest visual acuity of horses, which despite their large eyes, is limited by factors such as low retinal ganglion cell density and the narrow visual streak. Thus, while CT morphometry provides precise anatomical insights, it also highlights inherent functional constraints of the equine visual system. Further studies including live animals would allow the assessment of physiological parameters such as intraocular pressure, vascular dynamics, and potential age-related changes that cannot be evaluated in cadaveric material. In addition, expanding the sample to include different breeds, broader age groups (particularly juveniles and geriatrics), and clinical cases will be essential to refine normative data, improve understanding of anatomical and functional variability, and enhance the clinical applicability of these findings in diverse equine populations.

## Figures and Tables

**Figure 1 animals-15-03165-f001:**
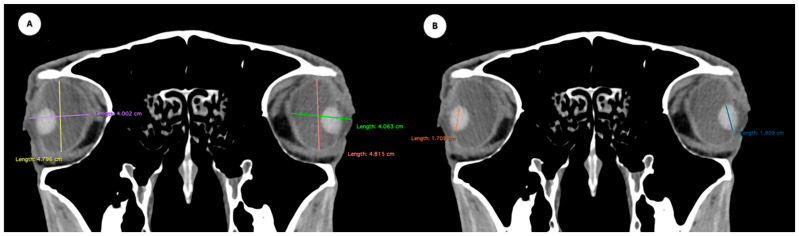
(**A**) Transverse multiplanar reconstruction (MPR) of the equine eyeball showing the maximum anterior–posterior distance (axial length), measured from the external surface of the cornea to the internal surface of choroid/retina/sclera, and the maximum latero-medial distance (equatorial width), measured perpendicular to the axial length. (**B**) Transverse MPR illustrating lens dimensions, with the latero-medial distance (equatorial width), measured at its widest point.

**Figure 2 animals-15-03165-f002:**
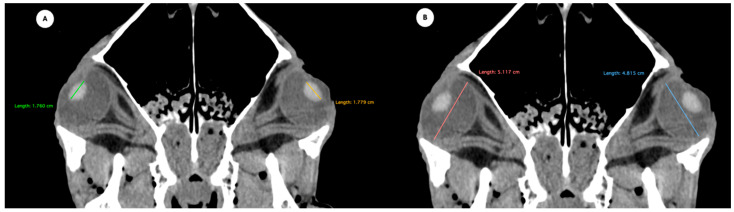
Dorsal multiplanar reconstruction (MPR) images showing (**A**) the maximal latero-medial distance of the lens and (**B**) the maximal latero-medial distance of the eyeball, measured perpendicular to the axial length, immediately caudal to the lens.

**Figure 3 animals-15-03165-f003:**
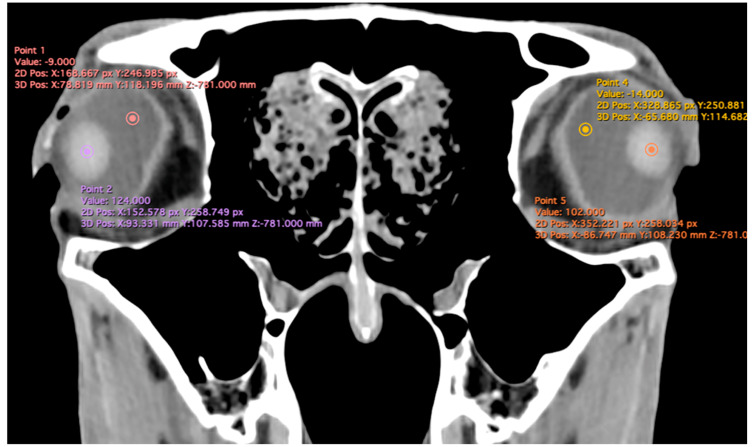
Transverse multiplanar reconstruction (MPR) illustrating attenuation values obtained by placing ROIs centrally within each structure to measure the densities of the anterior chamber, posterior chamber, and lens.

**Figure 6 animals-15-03165-f006:**
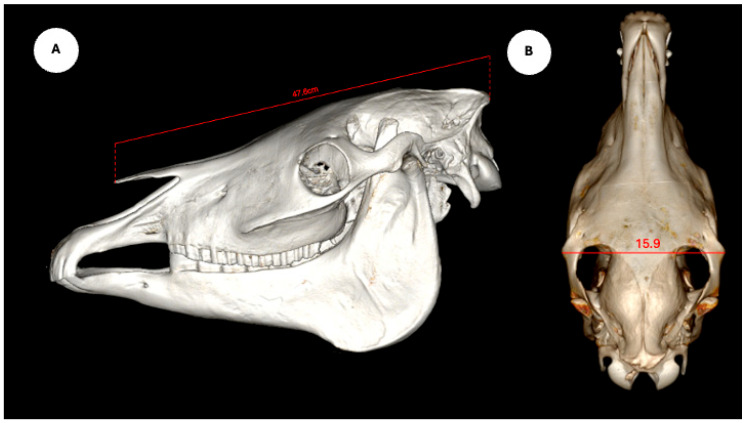
Volume rendered images of the equine skull showing (**A**) skull length, measured from the external occipital protuberance to the rostral border of the nasal bone, and (**B**) skull width, measured between the lateral edges of the zygomatic arches.

**Table 1 animals-15-03165-t001:** Measurements of the right and left eye.

	Right Eye				Left Eye				Both Eyes			
	Mean	Median	Range	sd	Mean	Median	Range	sd	Mean	Median	Range	sd
Ocular bulb height (cm)	5.19	5.20	(5–5.5)	0.14	5.19	5.20	(5–5.4)	0.11	5.19	5.19	(5–5.5)	0.13
Ocular bulb width (cm)	4.29	4.25	(4.1–4.6)	0.14	4.27	4.27	(4.1–4.47)	0.10	4.28	4.26	(4.1–4.6)	0.12
Ocular bulb length (cm)	5.09	5.08	(4.86–5.4)	0.16	5.09	5.10	(4.78–5.32)	0.18	5.09	5.1	(4.78–5.4)	0.17
Orbital cavity height (cm)	5.27	5.23	(5.1–5.5)	0.10	5.28	5.27	(5–5.46)	0.14	5.27	5.27	(5–5.5)	0.12
Orbital cavity length (cm)	6.29	6.28	(6.04–6.5)	0.13	6.29	6.30	(6–6.6)	0.15	6.29	6.3	(6–6.6)	0.14
Lens height (cm)	1.53	1.52	(1.47–1.66)	0.05	1.53	1.52	(1.4–1.63)	0.06	1.53	1.52	(1.4–1.66)	0.06
Lens width (cm)	1.24	1.24	(1.12–1.34)	0.05	1.22	1.22	(1.05–1.35)	0.07	1.23	1.23	(1.05–1.35)	0.06
Lens length (cm)	1.63	1.63	(1.52–1.69)	0.04	1.63	1.65	(1.5–1.69)	0.05	1.63	1.64	(1.5–1.69)	0.04
Posterior chamber width (cm)	2.24	2.25	(2.1–2.39)	0.09	2.28	2.29	(2.09–2.4)	0.09	2.26	2.26	(2.09–2.4)	0.09
Anterior chamber width (cm)	0.57	0.57	(0.42–0.67)	0.06	0.56	0.54	(0.4–0.69)	0.07	0.56	0.57	(0.4–0.69)	0.07
Posterior chamber (UH)	−4.62	−5.00	(−8–−2)	1.63	−3.81	−4.00	(−7–−2)	1.50	−4.21	−4	(−8–−2)	1.60
Anterior chamber (UH)	−8.57	−8.00	(−14–−6)	2.13	−7.48	−7.00	(−13–−4)	2.02	−8.02	−8	(−14–−4)	2.12
Lens (UH)	129.62	129.00	(113–147)	7.41	128.43	129.00	(101–139)	7.62	129.02	129	(101–147)	7.45

## Data Availability

The information is available at “https://accedacris.ulpgc.es”, accessed on 25 June 2025.

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
