# Peer review of "Computed Tomographic Assessment of Normal Ocular Dimensions and Densities in Cadaveric Horses (Equus ferus caballus)"

_animals, 2025, doi:10.3390/ani15213165_

Round 1

Reviewer 1 Report

Comments and Suggestions for Authors

Dear authors,

The study is well presented and explained. I do not have specific comments about it.

I would only suggest if you could include in the discussion some perspectives, as what are the next steps after this research? or what was the objective of the study in terms of further application of these results? what could come next?

------------------------

• What is the main question addressed by the research?

The study aimed to describe the computed tomographic (CT) dimensions and contrast attenuation characteristics of the eye in horses

• Do you consider the topic original or relevant to the field? Does it address a specific gap in the field? Please also explain why this is/ is not the case.

As I explained in my comments, it will be nice for the authors to explore the potential perspectives if the study, as for clinical applications or further research, as to foresee this problematic ( implants after enucleation, lens implants?)

• What does it add to the subject area compared with other published material?

According to my research, There are about 15 or so similar studies (i.e. PMID 30716192, 3817140, etc). This is why I answer in your review survey, that it is an original work as per the number of horses studied and the comparisons they make in their analysis, which differ from what it is in the literature. However, with low interest overall, as authors do not develop perspectives and the future of this pilot study; applicability in certain pathologies, further research? Contrast CT?

• What specific improvements should the authors consider regarding the methodology?

As I mentioned in the review, the manuscript and research work are well done and written, hence explained. In my opinion, their methodology is satisfactory.

• Are the conclusions consistent with the evidence and arguments presented and do they address the main question posed? Please also explain why this is/is not the case.

The conclusions and results are consistent.

Again, the research work is well developed, factually explained and well justified. The manuscript is of good quality. My only observation, as a surgeon and as a researcher, it is that it is hard for me to see the future applications of this work, hence its relevance and interest for the reader, point that shall be developed as I suggested it to the authors.

Author Response

Dear Reviewer,

We sincerely appreciate your thoughtful suggestions, as they help us to better define the objectives of the study and the potential applications of its results. In response, we have included the following paragraph in the Introduction and  Discussion sections:

"This study provides baseline CT reference values for ocular dimensions and radiodensities in horses, together with preliminary correlations with cranial morphometrics. Looking ahead, future research should focus on validating these findings in live animals, which would allow the evaluation of physiological parameters such as intraocular pressure, vascular perfusion, and age-related changes that cannot be reliably assessed in cadaveric specimens. Additionally, expanding the sample to include different breeds and age groups (particularly foals and geriatric horses), as well as clinical cases with ocular disease, would refine normative data and improve their applicability to diverse equine populations. From a clinical perspective, these morphometric data could contribute to the development of ocular prostheses after enucleation and the design of intraocular lens implants, where precise anatomical references are still scarce. Therefore, the results of this research not only advance anatomical knowledge but also open the door to innovative diagnostic and surgical applications in equine ophthalmology."

Reviewer 2 Report

Comments and Suggestions for Authors

The authors describe some morphometrical data of the eyes and ocular cavities of 20 horses by means of computed tomography. The manuscript appears intersting, well written and equipped by appropriate figures. The number of 20 individuals , i.e. 40 eyes, is appropriate and valid for statistical analyses. The introduction, material and methods are well described. The discussion plan and the references valid and updated. Overall this work is convincing and it can be useful for practical and clinical procedures. 

However there are some little problems that the authors could have solve to improve the quality of this manuscript and make it whorthy of publication:

i ) the authors should clarify how much time passed between death and the freezing of the heads. Indeed, as the authors well know, in Canarian Islands the temperatures are very high and can greatly affect  the preservation of soft tissues. 

ii) some information on the causes of death would be appreciated by the reader.

One major point regards the results and particularly Table 1. Indeed, for example, when I read the first line regarding the ocular bulb height, I see that the mean value of right eyes is 5.19 and in left eyes is 5.19. This means that the mean value of both eyes must be 5.19, and instead it is 5.06.  This happens for almost all values. The authors should explain  how they obteined these values, because at first glance they seem to have no mathematical logic. This is a point very important.

Moreover, in the discussion the authors should explain why one eye of a site is greater than the other site.

Author Response

Dear Reviewer,

We sincerely appreciate your comments and suggestions, which have been quite helpful in improving the manuscript.

Comment 1: the authors should clarify how much time passed between death and the freezing of the heads. Indeed, as the authors well know, in Canary Islands the temperatures are very high and can greatly affect  the preservation of soft tissues. 

Response: Thank you for this relevant comment. Following your suggestion, the following information has been added "Immediately after euthanasia, the carcasses were frozen in their entirety to preserve soft tissues and prevent autolytic changes. Once frozen, the heads were separated using an electric saw and stored under frozen conditions until imaging. This protocol was particularly important given the high ambient temperatures in the Canary Islands, where any delay between death and freezing could compromise tissue integrity".

Comment 2: some information on the causes of death would be appreciated by the reader.

Response: We thank the reviewer for this suggestion. We have now added information regarding the causes of euthanasia in the Materials and Methods section. The revised text reads as follows: “In this cohort, the most frequent reasons for euthanasia were colic, laminitis, and, less commonly, fractures.

Comment 3: One major point regards the results and particularly Table 1. Indeed, for example, when I read the first line regarding the ocular bulb height, I see that the mean value of right eyes is 5.19 and in left eyes is 5.19. This means that the mean value of both eyes must be 5.19, and instead it is 5.06.  This happens for almost all values. The authors should explain  how they obtained these values, because at first glance they seem to have no mathematical logic. This is a point very important.

Response: We thank the reviewer for carefully checking our results and for pointing out the inconsistency in Table 1. After revising the dataset, we realized that the issue was due to a copy-and-paste error when transcribing the values into the table. We confirm that this mistake only affected the presentation of the descriptive statistics in Table 1 and did not influence the calculations, analyses, or conclusions presented in the manuscript. We have now corrected the table and the results accordingly.

Comment 4: Moreover, in the discussion the authors should explain why one eye of a site is greater than the other site.

Response: The Mann–Whitney U test was used to compare measurements between the right and left eyes, but the results section does not report any statistically significant differences between them. This suggests that no eye was found to be consistent or significantly larger than the other. Therefore, we have added the following text in the results: “The Mann–Whitney U test revealed no statistically significant differences in the measurements between the right and left eyes. Similarly, when all variables were considered collectively, the analysis confirmed the absence of significant differences between both eyes.”

Reviewer 3 Report

Comments and Suggestions for Authors

The manuscript (animals-3825766) entitled “Computed Tomographic Assessment of Normal Ocular Dimensions and Densities in Cadaveric Horses (Equus ferus ca-ballus)” authored by Enara Lucas Parra, Lidia Pitti, Mario Encinoso, Ana Sofia Ramírez, Marcos Fumero Hernández, Maria Luisa Díaz-Bertrana *, Inmaculada Morales, Alberto Arencibia, and Jose Raduan Jaber aimed to characterize the CT features of the normal equine eye, establishing reference values ​​for the dimensions, volumes, and radiodensities of ocular structures, and to evaluate potential correlations between these parameters and skull measurements. The items described in the manuscript meet the scope of the journal Animals, in “Physiology and physical structure of animals”. Thirty-two bibliographic references were used, the oldest dating from 1992, but the majority from 2010 to the present. Several articles used as references contain the names of the manuscript's authors, but these articles use the same technique on animals of other species, appearing to be a continuation of research, including publications in Animals. I leave it to the editors' discretion whether there is a conflict of interest with self-promotion, but this reviewer does not see it that way. The journals where the cited articles were published are of high scientific importance, lending greater credibility to the article in question. The introduction conceptualizes and explores the topic clearly, but since it was quite short, a brief account of the anatomy of the eye and the importance of its structures is suggested, taking into account the equine species, with its particularities in relation to other species. These animals, unlike cattle, are used for sports and on rural properties for fieldwork, relying heavily on their vision. The spacing between paragraphs in the manuscript should be corrected. In lines 171 to 173, the authors justify the use of statistical tests, which is a good idea. The results are moderately well presented, though the discussion isn't particularly good, as it uses more of the authors' own articles for comparison and neglects the actual discussion of the results obtained in horses, compared to other studies and their practical applicability, which is very superficial. They should report on the impacts of diseases, such as ocular neoplasms, on ocular dimensions, but this is an anatomical study. I believe this would enrich the topic.

On line 66, there is a citation, formatting, and source error. Where it reads [5] it should be [6] and not in bold.
On line 75, correct it to Material and Methods.
In Material and Methods, the origin of these animals should be described. Were these animals treated by a specific service provider, or several? Were these animals dead of natural causes or euthanized on medical advice? Was an ophthalmic evaluation performed on the cadavers? If so, how long after death? This is important, given ocular dehydration, autolysis, and other factors that may have altered the actual values ​​of the measurements while the animal was still alive. In Spain, does the fact that the animal died relieve researchers of the responsibility to prove the origin of the specimens? The authors report "in accordance with applicable laws," but do they know what those laws are?
On line 88, I believe this sentence should be included in the previous text, explaining how these animals were frozen and stored. Specify, for example, that when the animals died, they were donated for study and remained in a cold chamber at temperature "x" or "y" until the time of examination, and so on.
On lines 90 and 91, was only the head scanned? If so, were the external tissues preserved? If reflected, what was reflected? There are gaps in the description that need to be filled.
On line 107, insert a comma between "oblique" and "sagittal."
Authors should standardize the line spacing of captions.
During the construction of the manuscript, grammatical rules must be followed. This is where the work of the authors and designers should be included in the final product. Collaboration is essential for a good presentation of the material to the public.
The paragraph located on lines 162 to 169 can be rewritten to indicate the results in a different way. For example: "The average ocular measurements indicated that height is greater than width in the horses evaluated, indicating the oval shape of equine eyes." With this information, readers will look for confirmation in the table, prompting them to read further. Likewise, other readers will read the table and then look at the text to understand the observed results. In this way, the authors provide the same information from different perspectives, improving the interpretation of the results.
For the explanation between lines 174 and 179, I suggest the same methodology as previously indicated. Instead of including values ​​here in the description, insert a new table with the values, with their respective indicative letters, and construct the text simply explaining whether there was a correlation or not and whether it was positive or negative. One idea is to create a graph with the models explaining the relationship. That would be nice.
There are several citations in the body of the text that do not meet the journal's standards (lines 239, 262, 273, and 288).
In lines 258 and 259, the information loses reliability due to the number of animals cited, including only one juvenile. For more reliable data, at least the same number of young animals would be needed for the number of adults. Modify this statement to include a juvenile animal in the study, opening the door for further studies on animals with a wider age range.
In line 331, correct the punctuation.
Definitely, systematically review the Animals journal's publication guidelines for writing references, which are quite messy in formatting and missing information (reference 30 is missing the year).

Comments on the Quality of English Language

The English is almost adequate, but could use technical tweaks that would aid textual comprehension. A native-speaking proofreader could assist the authors with the writing, which would make the manuscript more polished and elegant.

Author Response

Dear Reviewer,

We are truly grateful for your thoughtful comments and constructive suggestions. They have been instrumental in enhancing the quality and clarity of our manuscript. The responses are listed below.

Comment: The introduction conceptualizes and explores the topic clearly, but since it was quite short, a brief account of the anatomy of the eye and the importance of its structures is suggested, taking into account the equine species, with its particularities in relation to other species. 

Response: We thank the reviewer for this helpful suggestion. A brief description of the equine eye anatomy and the functional importance of its main structures has been incorporated into the Introduction, highlighting the particularities of the equine visual system in comparison with other species. The revised text now reads as follows (lines 41–48): The equine eye is among the largest of all terrestrial mammals and plays a pivotal role in the daily activities of horses. Unlike cattle, horses are extensively used in sports, leisure, and fieldwork, relying heavily on vision for both performance and safety. Anatomically, the equine globe is distinguished by its considerable axial length and broad visual field—adaptations that facilitate early detection of predators and environmental hazards. However, these same structural features also pose clinical challenges, as the eye is particularly susceptible to trauma and ocular disease, which can markedly compromise vision and, in turn, the animal’s welfare and utility.

Comment: The spacing between paragraphs in the manuscript should be corrected. 

Response: The spacing between paragraphs  has been corrected.

Comment: the discussion isn't particularly good, as it uses more of the authors' own articles for comparison and neglects the actual discussion of the results obtained in horses, compared to other studies and their practical applicability, which is very superficial. They should report on the impacts of diseases, such as ocular neoplasms, on ocular dimensions, but this is an anatomical study. I believe this would enrich the topic.

Response: We appreciate the reviewer’s constructive feedback regarding the discussion section. In the revised version of the manuscript, the discussion has been substantially expanded to provide a deeper interpretation of our results in horses, emphasizing comparisons with previous studies in both equine and other species. Additional context has also been included to highlight the clinical and practical applicability of our findings, particularly in relation to conditions that may affect ocular dimensions—such as trauma, inflammation, and neoplastic processes.

Comment: On line 66, there is a citation, formatting, and source error. Where it reads [5] it should be [6] and not in bold.

Response: We really appreciate your carefully revision,  we have corrected  the citation as you recommended. 

Comment: On line 75, correct it to Material and Methods.

Response: It has been corrected.

Comment: In Material and Methods, the origin of these animals should be described. Were these animals treated by a specific service provider, or several?

Response: We appreciate the reviewer’s observation. The origin of the animals has been clarified in the revised version of the manuscript. All horses included in the study were treated at the Equine Medicine and Surgery Service of the Veterinary Teaching Hospital, University of Las Palmas de Gran Canaria. This information has been added to the Materials and Methods → Animals section.

Comment: Were these animals dead of natural causes or euthanized on medical advice? Was an ophthalmic evaluation performed on the cadavers? If so, how long after death? 

Response: All horses included in this study were euthanized for medical reasons unrelated to this research. The most frequent causes of euthanasia were colic, laminitis, and, less commonly, fractures. Regarding the ophthalmic evaluation, each animal underwent a complete examination by a certified ophthalmologist as part of its clinical assessment prior to euthanasia, and only those confirmed to be free of ocular abnormalities were included in the study. This information has added in the revised manuscript.

Comment: The authors report "in accordance with applicable laws," but do they know what those laws are?

Response: We appreciate the reviewer’s observation. The statement has been clarified in the revised manuscript to specify the applicable legal framework. The study was conducted in accordance with Spanish Royal Decree 53/2013, which regulates the protection of animals used for scientific purposes, and with the Directive 2010/63/EU of the European Parliament on the same subject. Since the research was carried out exclusively on cadaveric specimens, it did not require ethical approval under these regulations.

Comment: On line 88, I believe this sentence should be included in the previous text, explaining how these animals were frozen and stored. Specify, for example, that when the animals died, they were donated for study and remained in a cold chamber at temperature "x" or "y" until the time of examination, and so on.

Response: We appreciate the reviewer’s valuable observation. The suggested clarification has been incorporated into the revised manuscript to provide a more detailed description of the freezing and storage process. The revised text now reads as follows: Immediately after euthanasia, the carcasses were donated for research purposes and stored in a cold chamber at –20 °C to preserve soft tissues and prevent autolytic changes. Once completely frozen, the heads were separated using an electric saw and maintained under frozen conditions until imaging.”

Comment: On lines 90 and 91, was only the head scanned? If so, were the external tissues preserved? If reflected, what was reflected? There are gaps in the description that need to be filled.

Response: We thank the reviewer for this pertinent comment. Indeed, only the heads were scanned in this study. The external tissues, including skin, muscles, and eyelids, were fully preserved to maintain the natural anatomical relationships of the orbital region. This clarification has been added to the revised manuscript 

Comment: On line 107, insert a comma between "oblique" and "sagittal."

Response: It has been inserted.

Comment: Authors should standardize the line spacing of captions.

Response: As you recommend, we have standardized the line spacing of captions.

Comment: The paragraph located on lines 162 to 169 can be rewritten to indicate the results in a different way. For example: "The average ocular measurements indicated that height is greater than width in the horses evaluated, indicating the oval shape of equine eyes." With this information, readers will look for confirmation in the table, prompting them to read further. Likewise, other readers will read the table and then look at the text to understand the observed results. In this way, the authors provide the same information from different perspectives, improving the interpretation of the results.

Response: We thank the reviewer for this valuable suggestion. The paragraph describing the ocular measurements has been rewritten to present the results more clearly and interpretively, emphasizing the proportional relationship between height and width that defines the oval shape of the equine eye. This modification enhances the readability of the Results section and better connects the textual description with the data summarized in Table 1.

Comment: For the explanation between lines 174 and 179, I suggest the same methodology as previously indicated. Instead of including values ​​here in the description, insert a new table with the values, with their respective indicative letters, and construct the text simply explaining whether there was a correlation or not and whether it was positive or negative. One idea is to create a graph with the models explaining the relationship. That would be nice.

Response: We appreciate the reviewer’s constructive suggestion regarding the presentation of the correlation results. In the revised version of the manuscript, we have modified the textual explanation of the correlation outcomes to indicate the strength and direction (positive or negative) of each correlation.

While we acknowledge the reviewer’s idea of adding a graphical or tabular representation, we opted to maintain a concise textual description because only a few statistically significant associations were observed. This approach avoids redundancy while preserving the descriptive and comparative nature of the study. The revised paragraph clearly conveys whether the correlations observed were statistically significant and their respective directionality, thus improving the interpretability of the results as suggested.

Comment: There are several citations in the body of the text that do not meet the journal's standards (lines 239, 262, 273, and 288).

Response: Thank you for pointing this out. We have revised the citations on lines 239, 262, 273, and 288 to ensure they fully comply with the journal’s formatting standards.

Comment: In lines 258 and 259, the information loses reliability due to the number of animals cited, including only one juvenile. For more reliable data, at least the same number of young animals would be needed for the number of adults. Modify this statement to include a juvenile animal in the study, opening the door for further studies on animals with a wider age range.

Response: As  you recommend, we have modified the statement to open the door for further studies on animals with a wider age range (see lines 369-373).

Comment: In line 331, correct the punctuation.

Response: The punctuation has been added (now line 354).

Comment: Definitely, systematically review the Animals journal's publication guidelines for writing references, which are quite messy in formatting and missing information (reference 30 is missing the year).

Response: We apologize for the errors in the reference list. In this revised version, the references have been formatted in full accordance with the publication guidelines of Animals journal.

Finally, we appreciate the reviewer’s comment regarding the language quality of the manuscript. The entire text has been thoroughly revised by the authors with the assistance of a fluent English speaker experienced in scientific writing to improve clarity, fluency, and technical precision. Particular attention has been given to terminology consistency, sentence structure, and overall readability. We trust that the current version of the manuscript now meets the journal’s linguistic standards and provides a polished and professional presentation.